# Exploring the Pipeline of Novel Therapies for Inflammatory Bowel Disease; State of the Art Review

**DOI:** 10.3390/biomedicines11030747

**Published:** 2023-03-01

**Authors:** Yasmin Zurba, Beatriz Gros, Mohammad Shehab

**Affiliations:** 1Department of Internal Medicine, Mubarak Al-Kabeer Hospital, Kuwait University, Jabriya 46300, Kuwait; 2IBD Edinburgh Unit, Department of Medicine, Western General Hospital, Edinburgh EH4 2XU, UK; 3Gastroenterology Department, Reina Sofía University Hospital, 30003 Córdoba, Spain

**Keywords:** Crohn’s disease, ulcerative colitis, Janus kinase inhibitor, interleukins, IBD, small molecules

## Abstract

Crohn’s disease (CD) and ulcerative colitis (UC), known as inflammatory bowel diseases (IBD), are characterized by chronic inflammation of the gastrointestinal tract. Over the last two decades, numerous medications have been developed and repurposed to induce and maintain remission in IBD patients. Despite the approval of multiple drugs, the major recurring issues continue to be primary non-response and secondary loss of response, as well as short- and long-term adverse events. Most clinical trials show percentages of response under 60%, possibly as a consequence of strict inclusion criteria and definitions of response. That is why these percentages appear to be more optimistic in real-life studies. A therapeutic ceiling has been used as a term to define this invisible bar that has not been crossed by any drug yet. This review highlights novel therapeutic target agents in phases II and III of development, such as sphingosine-1-phosphate receptor modulators, selective Janus kinase inhibitors, anti-interleukins, and other small molecules that are currently under research until 1 January 2023. Emerging treatments for CD and UC that have just received approval or are undergoing phase III clinical trials are also discussed in this review.

## 1. Introduction

Inflammatory bowel diseases (IBD) are inflammatory chronic-relapsing conditions whose major forms are Crohn’s disease (CD) and ulcerative colitis (UC). These diseases are also known for their complications, which occur due to chronic inflammation in most cases, leading to an increased risk of hospitalization, surgery, colorectal cancer, and disability with a high impact on their quality of life. Due to the persistent and progressive nature of IBD, effective treatment should be started ideally at the early stages of the disease to avoid relapses and complications [1]. This would reduce the potential use of corticosteroids, the need for surgery and hospital admission, and improve the overall well-being of IBD patients [2]. This burden on health-care resources is further influenced by a rising global IBD prevalence rate over the years [3]. The Selecting Therapeutic Targets in Inflammatory Bowel Disease (STRIDE) program, recommends targets used for a treatment strategy of treat-to-target. STRIDE 2 recommends the same clinical and endoscopic healing targets mentioned in STRIDE 1 with the addition of serum and fecal biomarker normalization, quality of life restoration, disability prevention, and normal growth in children [4,5]. The current IBD therapeutic armamentarium is confirmed by 5-aminosalicylic acid (5-ASA) compounds, immunosuppressants such as methotrexate and thiopurine, anti-tumor necrosis factor (TNF) alpha, anti-integrin, and anti-IL-12 and 23 among the monoclonal antibodies. Recently, tofacitinib, filgotinib, upadacitinib, and ozanimod, which are non-biological small molecules, have been approved for the treatment of moderate-to-severe UC [2,6].

The purpose of this review article is to describe novel and emerging therapies in IBD. It will highlight the new and emerging target agents such as selective JAK inhibitors, anti-ILs, TNF inhibitors (like polyclonal antibodies), anti-adhesion molecules (including monoclonal antibodies), sphingosine-1-phosphate (S1P) receptor modulators (immunomodulators) along with small molecules (see Table 1). Some other small molecules that are in the development phase have also been reviewed; other treatment strategies, like stem cell and microbiome targeted treatment, are not included in this review.

## 2. Targets for Small Molecules and Biologics

Cytokines, including interleukins (ILs) and interferons (IFNs), play an important role in inflammatory responses. When cytokines are released by antigen-presenting cells (APCs), such as dendritic cells in the intestines, they trigger an adaptive immune response through the differentiation of several T cells. Studies have shown that IL-12, 22, and 23 are involved in IBD pathogenesis. It is suggested that IL-12 is involved in the initiation of intestinal inflammation caused by epithelial barrier disruptions [7]. Moreover, IL-12 works together with IL-23 to maintain chronicity, with IL-12 being more prominent in the earlier stages and IL-23 in the later stages. IL-23 is also involved in the differentiation of naive T helper (Th) 17 cells, which increase the secretion of other inflammatory cytokines like IL-17 and IL-22 [8]. On the other hand, IL-12, has been found to induce Th1 polarization, which is further associated with IFN-γ and tumor necrosis factor (TNF) production. Therefore, IL-12 and IL-23 play a crucial role in the maintenance of inflammation in IBD. Different studies have determined that targeting IL-12 also impacts its downstream cytokines, ultimately resulting in the control of IBD activity. These results were observed in different studies on colitis conducted on both animals and humans [9,10]. The first anti-IL drug developed and approved for IBD was ustekinumab.

Similarly, increased levels of IL-23 have been observed in several colitis disease models, like T-cell transfer colitis, Helicobacter hepaticus colitis, 2,4,6-trinitrobenzene sulfonic acid solution (TNBS) colitis, and dextran sodium sulfate colitis [11]. IL-23 is involved in JAK2 and Tyrosine Kinase (TYK) 2 activation, ultimately leading to STAT3 and STAT4 production. The role of IL-23 in CD has been supported through different genome-wide association studies (GWAS) [12], where a specific mutation helped impart resistance against CD in more than 75,000 cases [13].

IL-22 is considered to have both protective and deleterious effects on intestinal inflammation. It is secreted from various cells in the intestines, including CD4+ T cells, CD8+ T cells, neutrophils, and dendritic cells (DCs). IL-22 can activate STAT3 in a constitutive manner, which is a major player in the pathogenesis of IBD [14]. JAK/STAT signaling is involved in several pathways that are disturbed in IBD because of an imbalance in the population of T cells. In UC patients, there is Th2-mediated inflammation, while an increase in Th1 cells mostly characterizes CD. Th1 cells activate IL-12 and STAT4 pathways via JAK2 and TYK2. Similarly, IFN-γ is also associated with the differentiation of Th1 cells, for which phosphorylation of STAT1 by JAK1 and 2 play a major role [15].

Moreover, JAK/STAT signaling is involved in pro-inflammatory responses in intestinal epithelial cells (IECs) and myeloid cells. For example, elevated STAT3 levels are reported in different cases of IBD [16], where it was found that the mu-opioid receptor agonist, DALDA, promotes the proliferation of IECs while depending on STAT3. Therefore, DALDA has a protective effect on the intestine through a mechanism that involves activation of STAT3.

Because ILs, IFN-γ, and JAK are directly involved in inflammatory responses, targeting their pathways seems to be a promising therapy for IBD. For this reason, several biologics and small molecules have been developed and tested. Biologics have been designed to control the effects of different cytokines. Moreover, many of the small molecule inhibitors designed to inhibit JAK are also designed to inhibit JAK3, as one of the biggest interests regarding these molecules is to decrease their possible side effects [17]. JAK3 is mostly expressed in hematopoietic cells and signals via the IL-2R family. On the other hand, there is a lack of long-term safety data available for JAK1 and JAK2 inhibitors, although some safety signals have been observed in animal models [18].

The newest class of oral small molecules approved by the FDA, sphingosine-1 phosphate (S1P) receptor–modulating therapies, show promising results. S1P is a bioactive lipid mediator that functions by activating the cell-surface G protein-coupled receptors S1P1–S1P5. S1P1 is the most ubiquitous of the S1P receptors and is found in both lymphocytes and endothelial cells. When S1P binds to S1P1, it is taken intracellularly, and in turn, the cell surface agonist does not signal. This results in fewer circulating lymphocytes and a decreased inflammatory reaction. Due to this mechanism, it is being used and further studied for the treatment of IBDs [19]. Figure 1 and Table 1 show clinical trials of biological and small molecules for the treatment of IBD.

## 3. Biologics and Small Molecules

### 3.1. IL Inhibitors

#### 3.1.1. Selective Inhibitors of IL-23

Monoclonal antibodies that specifically target IL-23 (subunit p19) include brazikumab, risankizumab, mirikizumab, and guselkumab.

#### 3.1.2. Risankizumab

Randomized phase III trials, ADVANCE and MOTIVATE, have evaluated the safety and efficacy of risankizumab for moderate-to-severe CD [20]. These showed significantly greater rates of complete remission in moderate-to-severe CD at 12 weeks [21,22]. Moreover, treated patients who did not show complete remission after 12 weeks of administration of risankizumab were further subjected to open-label IV risankizumab administration, which showed benefit. Data on maintenance has also recently been published in the FORTIFY phase III study [23]. Recently completed clinical phase III trials also reported the effectiveness and tolerance of risankizumab in this setting [21]. No safety issues were found in these studies, including the phase II open-label study [24,25]. Thus, risankizumab has been proven to be an effective therapy to induce and maintain remission in CD.

#### 3.1.3. Brazikumab

An OL phase IIa clinical trial of brazikumab in patients with moderate to severe CD was conducted, consisting of a 12 week induction period with 700 mg of IV brazikumab or placebo followed by 210 mg of subcutaneous brazikumab every four weeks. Two brazikumab injections at weeks zero and four resulted in a greater rate of therapeutic outcome at week eight when compared to the placebo. Further SC dosing every four weeks resulted in a steady treatment outcome. It should be noted that this phase IIa trial did not include any imaging or endoscopic examination [26]. Moreover, after a follow-up period of up to 100 weeks, brazikumab was reported as generally safe and well tolerated in this study. More clinical trials that study the efficacy of brazikumab for the treatment of CD and UC are in progress (Table 1).

#### 3.1.4. Guselkumab

Guselkumab is a monoclonal human IgG1 antibody currently approved in the United States, Canada, Japan, and the European Union for the treatment of moderate to severe psoriasis [27]. Phase II and III trials of guselkumab, GALAXI 1, GALAXI 2, and GALAXI 3 are being conducted on patients with CD and UC [28]. In these studies, pharmacokinetics, biomarkers, efficacy, and safety were evaluated. In GALAXI 1, the effectiveness and safety of guselkumab were evaluated in patients with moderate-to-severe CD who showed no response or an inadequate response to treatment with corticosteroids or immunosuppressive agents. Patients who received guselkumab as compared to a placebo presented better response, recovery, clinical biomarker response, and endoscopic improvement as compared to a placebo [29]. The drug was well tolerated across all studies.

#### 3.1.5. Mirikizumab

Mirikizumab is a humanized IgG4 monoclonal antibody. A recent phase II clinical trial carried out on patients with UC revealed that mirikizumab induced a clinical response after 12 weeks and was efficacious as a maintenance treatment for the disease [30]. It can be administered as an IV or SC formulation. In addition, an extended phase II study for an additional 12 weeks showed that 50.0% of initial non-responders could gain a response to mirikizumab and maintain this clinical response for up to 52 weeks with no additional safety concerns [31].

### 3.2. TNF Inhibitors

#### 3.2.1. AVX-470

AVX-470 is an orally administered, bovine-derived medication. It is an oral polyclonal anti-TNF antibody that acts by locally neutralizing TNF in the gastrointestinal system. In a first human, double blind, placebo-controlled, ascending and repeat-dose clinical trial published in 2016, 37 patients with active UC were randomly assigned to one of two treatment and placebo groups [32]. The treatment group was further divided into three ascending-dose cohort groups that received 0.2 g/day, 1.6 g/day, and 3.5 g/day. Disease activity was assessed by a colonoscopy performed before and after treatment or a placebo. The primary endpoint of this study was the safety profile of AVX-470, which proved to be well tolerated with no incidence of allergic reactions or opportunistic infections. The safety profile was similar across the different doses administered, with 52% of patients developing one or more adverse effects, whereas 78% of patients experienced adverse effects in the placebo group. After four weeks of treatment, 25.9% of patients achieved a clinical response compared with 11.1% in the placebo group, and the greatest improvements were reported in the high dose group (3.5 g/day). Therefore, AVX-470 was well tolerated and effective with regards to clinical, serum biomarker, and endoscopic outcomes in the treatment of UC.

#### 3.2.2. OPRX-106

Another orally administered TNF inhibitor is OPRX-106. It is a fused form of human anti-TNF receptor II and an IgG1 Fc domain expressed in plant cells, which function as a vehicle for delivery. A phase II randomized, two-arm, open-label clinical study included 25 patients with mild-to-moderate UC. Patients were randomized to receive 2 or 8 mg/day of OPRX-106 for eight weeks. Of the 25 patients enrolled in the trial, 18 completed it. Treatment response was obtained in 67.0% of cases, while remission was achieved in only 28.0% of patients. There were no anti-drug antibodies found with regards to toxicity. The study concluded that oral OPRX-106 shows beneficial biological activity and is well tolerated with no serious adverse effects [33].

### 3.3. Anti-Adhesion Molecules

#### 3.3.1. Etrolizumab

Unlike vedolizumab, the first anti-adhesion biologic drug used in IBD, which solely targets the α4β7 integrin, etrolizumab is a monoclonal anti-integrin antibody that has been humanized and specifically binds to the β7 subunit of both α4β7 and αEβ7 integrins. As a result, etrolizumab regulates inflammatory cell migration to the intestinal system and modulates its actions on the intestinal epithelium [34]. A phase III clinical trial and two OLE clinical trials are evaluating etrolizumab as a potential therapy for IBD [29]. The phase III clinical program, which has enrolled over 3000 patients, is studying etrolizumab as both an induction and maintenance therapy and includes the trials HIBISCUS I, HIBISCUS II, and GARDENIA [35,36,37]. As per the latest data from a phase III clinical trial, treatment with etrolizumab led to a higher response in comparison to a placebo in individuals with moderate-to-severe UC but failed to achieve its objective as a maintenance medication [38].

Moreover, in a phase II placebo-controlled study of etrolizumab, data was collected retrospectively from UC patients and controls. Gene expression profiling suggests that some individuals may show better responses to etrolizumab when GZMA and ITGAE mRNA genes are present in colon biopsies. Additionally, according to a recent study conducted on CD patients, etrolizumab led to a decrease in the activation of proinflammatory genes and cytotoxic intraepithelial lymphocytes (IEL) gene expression [39]. The safety profile was accepted and similar across all treatment groups [34].

#### 3.3.2. PN-943

PN-943 is an orally administered α4β7 antagonist that acts locally in the gastrointestinal system [40]. In the literature, PN-943 is compared to PTG-100, as they are both orally administered and gut-restricted α4β7 antagonist peptides. In two clinical trials conducted on mice comparing these medications as induction therapy for UC, PTG-100 was deemed less efficacious compared to PN-943 [41,42]. Furthermore, a phase II clinical trial for PTG-100 was discontinued, and another one is currently underway to evaluate the effectiveness of PN-943 as an induction and maintenance therapy with a 12 and 52 week follow-up, respectively, in moderate-to-severe UC patients [43]. Two studies conducted on healthy volunteers concluded that the drug has low systemic exposure with a safe pharmacological profile [40].

#### 3.3.3. PF-00547659

PF-00547659 is a human monoclonal antibody that reduces gastrointestinal inflammation by binding to a mucosal cell adhesion molecule (MadCAM), which is an adhesion receptor expressed by endothelial cells of lymphoid tissue in the intestine. PF-00547659 binds to MAdCAM to reduce lymphocyte migration to the gut. A phase II clinical trial was conducted evaluating the dose, efficacy, and safety of PF-00547659 on patients with moderate-to-severe CD who failed treatment with or were intolerant to immunosuppressive or TNF inhibitor agents. The results of this trial showed no significant decrease in the Crohn’s Disease Activity Index (CDAI) between the PF-00547659 and placebo-treated groups after 8 and 12 weeks [44]. Another study, TURNADOT, evaluated its effect on patients with moderate-to-severe UC [45]. The drug was given in three different doses: 7.5 mg, 22.5 mg, and 75 mg. The study concluded that, in this population, the medication was more efficacious than the placebo in inducing remission. Remission was defined as a Mayo score of ≤2 with no individual subscore <1 and a rectal bleeding score ≤1. The best effects were produced by PF-00547659 at doses of 22.5 mg and 75 mg. Furthermore, PF-00547659 was well tolerated and had a good safety profile.

#### 3.3.4. Abrilumab

Abrilumab is a monoclonal antibody that selectively targets α4β7 integrin, which is already a validated target in the treatment of IBD. Several clinical trials evaluated abrilumab for the treatment of moderate-to-severe CD and moderate-to-severe UC in patients with inadequate or loss of response to immunosuppressive agents, corticosteroids, or anti-TNF agents. In the study conducted on patients with CD, the primary endpoints of response at week eight and remission at week twelve were not met, although beneficial effects of the medication were observed for response and remission rates [46]. In contrast, in the clinical trial that enrolled patients with UC, the primary endpoint, remission at week eight, was achieved, along with clinical response and mucosal healing [47]. There were no major safety implications of abrilumab in either study compared to the placebo.

#### 3.3.5. Carotegrast Methyl (AJM300)

A small molecule inhibitor of α4 integrin, AJM300, has proven its efficacy against α4β7 and α4β1 receptors [48]. A phase IIa, double-blind, placebo-controlled trial involving 102 individuals with moderate-to-severe UC who failed or did not tolerate previous therapies, found significant rates of clinical remission, clinical response, and mucous membrane recovery in the treated group at eight weeks compared to the placebo group [49]. Despite only being reported in abstract form, the findings of a similar small CD trial in which substantially lower doses of AJM300 were administered to the study population revealed no significant difference in CDAI at week four between AJM300 and the placebo [50]. The drug is typically administered orally, three times per day, with good tolerance, and no serious adverse events were found in any of the clinical trials [51].

### 3.4. Sphingosine-1-Phosphate (S1P) Receptor Modulators

#### 3.4.1. Fingolimod

The S1P inflammatory target is known in multiple sclerosis, where fingolimod, a non-selective S1P receptor agonist, has been used since 2010 [52]. In IBD, the experience regarding this medication dates back to 2018. However, because of the related adverse reactions, more selective oral treatments, such as ozanimod and etrasimod, have been established for the effective treatment of IBD [53].

#### 3.4.2. Ozanimod

An oral selective immunomodulatory agonist for S1P1 and S1P5 receptors, ozanimod, has been approved for the treatment of relapsing forms of MS [54]. Supported by evidence from TOUCHSTONE, a phase II clinical trial, and TRUE NORTH, a phase III clinical trial, ozanimod was approved for the treatment of UC patients with moderate-to-severe disease [55,56].

TOUCHSTONE randomized patients to receive two doses of ozanimod, 0.5 mg per day and 1 mg per day, to assess clinical remission at eight weeks. The results showed that the 1 mg dose presented a higher rate of remission and mucosal healing for these patients. The medication was reported as both efficacious and safe during the open label extension (OLE) period of this study [57].

The results of TRUE NORTH, a double-blind, placebo-controlled phase III trial that studied ozanimod as an induction and maintenance therapy for moderate-to-severe UC. In this study, 645 patients were randomized to receive 1 mg per day of ozanimod or a placebo. Initially, after 10 weeks of therapy, 18.4% of study participants reached clinical remission, compared to 6.0% of those who received the placebo (*p* = 0.001). Of those who responded to this induction therapy, 37.0% maintained clinical remission after 52 weeks, vs. 18.5% in the placebo arm (*p* < 0.001) [58]. No new safety signals were reported in this clinical trial.

Moreover, when it comes to the use of ozanimod on CD patients, preliminary data from the STEPSTONE trial indicate the effectiveness of the medication, with 56.1% of patients who received the drug having a clinical response and 23.2% having an endoscopic response [58]. The safety profile of ozanimod in this study is thus far encouraging. Another placebo-controlled phase III trial is underway [59].

#### 3.4.3. Etrasimod

Etrasimod is a selective oral agonist for the S1P1, S1P4, and S1P5 receptors. A total of 156 patients with moderate-to-severe UC were randomly assigned to receive one of the two different doses of etrasimod, 1 mg or 2 mg, or a placebo once daily for 12 weeks in a double-blinded phase II trial [60]. After 12 weeks, the patients in the treatment arm, particularly those who received the higher dose of 2 mg, met all primary and secondary endpoints. Etrasimod at a dose of 2 mg had a significant improvement in the modified Mayo Clinic score when compared to the placebo (0.99 point difference). The endoscopic mucosal healing was 41.8% in the group receiving etrasimod 2 mg and 17.8% in the placebo group. In the treatment groups, treatment-related adverse events were mild-to-moderate. The most common adverse events included worsening of UC disease activity, nasopharyngitis, an upper respiratory tract infection, and anemia. Due to the aforementioned adverse effects, there were higher rates of discontinuation of the drug in the treatment arm vs. the placebo arm.

#### 3.4.4. Amiselimod

Amiselimod is an oral drug that modulates the S1P1 receptor. The results of a phase II trial, which randomized patients with moderate-to-severe CD into receiving 0.4 mg of amiselimod or a placebo, were published in 2021 and then updated in early 2022 [61]. Although treatment with amiselimod was considered well tolerated, 0.4 mg of amiselimod for 12 weeks did not differ significantly in terms of clinical response from receiving the placebo. Among the treatment group, 71.8% of patients completed the 14 week treatment period; seven participants had serious adverse events, including infections and cardiac disorders, and four discontinued the drug most likely for this reason.

### 3.5. Phosphodiesterase 4 Inhibitors

Phosphodiesterases (PDEs) are a group of enzymes that catalyze cyclic guanosine monophosphate (cGMP) and cyclic adenosine monophosphate (cAMP) breakdown. PDE4 is of particular interest as it causes cAMP breakdown, leading to the activation of the nuclear factor kappa B (NF-κB). NF-κB is responsible for the upregulation of proinflammatory cytokines [62].

#### Apremilast

Apremilast is an oral inhibitor of PDE4, which, when blocked, leads to the accumulation of cAMP. It has already been approved for the treatment of psoriasis and psoriatic arthritis [63]. This drug was tested in a double-blind, placebo-controlled phase II trial conducted on patients with active UC for three months who were naïve to, failed, could not tolerate, or had contraindications to conventional therapy [64]. Participants were randomized into three groups: 30 mg dose, 40 mg dose, and a placebo. The results showed that 31.6% of patients who received 30 mg of apremilast achieved the endpoint of clinical remission, compared to 12.1% of patients who received the placebo. Additionally, 73.7% of patients in the 30 mg group achieved endoscopic improvement, compared to 41.4% in the placebo group. Both the 30 mg and 40 mg groups experienced a decrease in serum CRP and fecal calprotectin compared to the placebo group. In conclusion, the primary endpoints of clinical remission were not met in this study; however, groups that received apremilast still experienced a level of improvement in clinical response, endoscopic features, and inflammatory markers. Additionally, remission was maintained in 40% of patients who continued taking the medication for up to 52 weeks. The safety analysis indicated 21.1% of patients in the apremilast group reported headaches, compared to 6.9% in the control group. Moreover, the serious adverse effect rate was reported to be 2.4% in the placebo group and 1.8% in the 40 mg dosage group.

### 3.6. Janus Kinase Inhibitors (JAKi)

#### 3.6.1. Tofacitinib

An oral JAK inhibitor, tofacitinib, was approved by the Food and Drug Administration (FDA) in 2018 for the treatment of UC. Tofacitinib inhibits the phosphorylation of JAK and its activation by taking away its ability to bind with ATP [65]. Despite receiving FDA approval for the management of UC, tofacitinib testing for CD was stopped after phase II trials after failing to meet primary endpoints.

In the pivotal OCTAVE 1, 2, and 3 trials, tofacitinib was deemed efficacious in the induction and maintenance of remission in UC patients with moderate-to-severe disease [66]. In a phase III trial, the effect of tofacitinib was evaluated based on the mean Inflammatory Bowel Disease Questionnaire (IBDF) score. The results indicated that the mean change from baseline was higher for tofacitinib in comparison to the placebo in both the induction and maintenance studies. The results also showed that 10 mg of tofacitinib given twice daily was significantly more efficacious in the induction phase of the study, whereas 5 mg twice daily was sufficient in the maintenance phase [67]. Overall, tofacitinib is well tolerated with a good safety profile, and adverse effects are similar to those observed with other drugs in the same population [68,69].

#### 3.6.2. Filgotinib

Filgotinib is a small molecule oral JAK1 phosphorylation inhibitor, approved in Europe in 2020 for rheumatoid arthritis [70]. It has also been evaluated for use in CD (the FITZROY study) and UC (the SELECTION study) [71,72]. In FITZROY, a double-blinded, placebo-controlled, randomized phase-II trial conducted on CD patients, the group that received filgotinib showed 47% clinical remission as compared to 23% in the placebo group. The primary endpoint was clinical remission, which was defined by a CDAI score less than 150 after 10 weeks of treatment. Moreover, safety data indicated that 9% of patients receiving the drug experienced adverse effects, compared to 4% in the placebo group, yielding no differences regarding safety [71].

The results of the phase IIb/III trial, the SELECTION trial, proved filgotinib’s efficacy in UC [72]. The study included patients who are biological therapy naïve and those who have previously received biological therapy. Each group was further divided into receiving filgotinib 100 mg/day, filgotinib 200 mg/day, or a placebo. Primary endpoints were endoscopic, rectal bleeding, and stool frequency (EBS) remission. Higher rates of clinical remission were observed in the 200 mg/day group (11.5% in the patients who had been exposed to biological therapies and 26.1% in those who had not) [65]. Patients who achieved clinical remission in the SELECTION trial were recruited to participate in a maintenance study [72] proving that filgotinib 100 mg/day and 200 mg/day were effective as maintenance therapy in patients with moderate-to-severe UC. Both groups met primary endpoints; however, only the 200 mg/day group met all secondary endpoints. Filgotinib was well tolerated in all study participants.

Further randomized trials of filgotinib in CD-DIVERSITY (phase III) and in DIVERGENCE (phase II) are currently in progress (NCT02914561 and NCT02914600).

#### 3.6.3. Upadacitinib

Upadacitinib received FDA approval in March 2022 for its use in the treatment of UC [73]. It is an oral and highly selective JAK1 inhibitor.

In a series of multicenter phase III trials, two replicate induction studies (U-ACHIEVE induction and U-ACCOMPLISH induction) and a single maintenance study (U-ACHIEVE maintenance) proved the efficacy and safety of upadacitinib in moderate-to-severe UC. Study participants were assigned to two cohorts for the induction study and to three cohorts for the maintenance study. In the two induction studies, the upadacitinib 45 mg groups had higher remission rates after 8 weeks of treatment compared to the placebo group (26% and 34% in the treated groups versus 5% and 4% in the placebo groups). In the maintenance study, patients were divided into three groups: upadacitinib 15 mg, upadacitinib 30 mg, or a placebo, and followed for 52 weeks. The upadacitinib groups had better remission rates at 52 weeks compared to the placebo group (42% in the group receiving 15 mg, 52% in the group receiving 30 mg, and 12% in the group receiving the placebo). When it comes to the safety of the drug, serious adverse events were reported more frequently in the placebo group when compared to the treatment group [74].

In addition, upadacitinib proved in a phase II trial to be effective in inducing endoscopic remission in patients receiving upadacitinib at dosages of 12 mg (22.0%) and 24 mg (14.0%), compared to the placebo (11%) [75]. Although the risk of adverse events was higher in the treatment group vs. the placebo group, the benefits outweighed the risks, and further trials on upadacitinib as a treatment for CD are supported. Upadacitinib phase III trials in moderate-to-severe CD have not yet been published, although some data from the U-EXCEL and U-EXCEED induction trials as well as the U-ENDURE maintenance trial have already been presented in recent international congresses. Phase III studies for the use of upadacitinib in the treatment of CD (NCT03345836 and NCT03345823) and UC (NCT03653026, NCT03006068, and NCT02819635) are currently underway. It is of note that the doses used in the phase II trials differ from those used in the phase III trials [76].

#### 3.6.4. Peficitinib

Peficitinib is an orally administered pan-JAK inhibitor with moderate selectivity for JAK3. This drug was approved for the treatment of moderate-to-severe rheumatoid arthritis in Japan in 2019, but trials involving its use in IBD management are still underway [77]. In a phase IIb trial to study the effectiveness and safety of peficitinib in patients with UC, participants were divided into five groups: placebo, once-daily peficitinib at doses of 25 mg, 75 mg, 150 mg, and twice-daily at the dose of 75 mg [78]. The results indicated that a dose-dependent response was insignificant in induction studies; however, groups receiving 75 mg per day or more of the drug showed significant remission rates when compared to the placebo group. While the drug was efficacious, safety analysis indicated that 45.5% of the patients treated with peficitinib exhibited adverse effects, compared to 34.9% in the placebo group. The leading adverse effects were anemia, worsening UC, and elevated blood creatinine phosphokinase levels.

#### 3.6.5. Izencitinib (TD-1473)

TD-1473, also known as izencitinib, is a gut-specific, oral, pan-JAK inhibitor [79]. Due to its gut-specificity, izencitinib exhibits limited systemic toxicity, and it proved to be effective in mouse models with minimal plasma concentrations [80]. In a first-in-human phase 1b clinical trial in moderate-to-severe active UC, patients were randomized to receive izencitinib 20 mg, 80 mg, 270 mg, or placebo for a total of 28 days. The results revealed that the drug was generally well tolerated, with a clinical response rate of 55% among the izencitinib-treated group at 270 mg compared to 11% in the placebo group. Additionally, compared to the placebo group, results in the treatment group showed an improvement in endoscopic disease activity (20% in the 20 mg group, 20% in the 80 mg group, and 9% in the 270 mg group) [81]. However, in a recent phase IIb trial to find the appropriate induction dose for patients with moderate-to-severe UC, izencitinib showed failed to achieve its primary endpoint of targeting a change in the total Mayo score at 8 weeks [82]. For the total duration of the study, all doses of izencitinib were well tolerated.

### 3.7. TYK2 Inhibitors

TYK2 is a member of the JAK-STAT family and is involved in intracellular cytokine signaling. TYK2 is also known to enhance interferon production in macrophages [83]. Due to its role in immune response and inflammation, the inhibition of TYK2 is a potential treatment target for IBD management.

#### 3.7.1. Brepocitinib

Brepocitinib (PF-06700841), a TYK2 and JAK1 inhibitor, has been clinically promising and well tolerated in the phase I and IIa trials for the treatment of psoriasis [84], and a phase III trial is currently in progress. Currently, phase II trials are recruiting participants to study brepocitinib as a treatment for CD [85]. VIBRATO, an umbrella study evaluating the efficacy and safety of brepocitinib and ritlecitinib, which are JAK3 and TEC inhibitors, indicated that both drugs are effective in patients with moderate-to-severe active UC compared to a placebo, as they met all the trial’s endpoints. These endpoints were described as total clinical remission, modified remission, or endoscopic improvement in patients with moderate-to-severe active UC compared to the placebo. The comparison of the efficacy of both drugs revealed that ritlecitinib had a higher rate of modified remission after 8 weeks. With regards to safety, brepocitinib was generally well tolerated.

#### 3.7.2. Deucravacitinib

Deucravacitinib is another TYK2 inhibitor that was effective in phase II and III trials in patients with plaque psoriasis and those with psoriatic arthritis [86]. In LATTICE-UC, a phase II double-blind trial to assess the efficacy and safety of deucravacitinib in patients with moderate-to-severe UC, both primary (clinical remission using the modified Mayo score) and secondary endpoints (clinical response using the modified Mayo score, endoscopic response, and histological improvement) were not met after 12 weeks [87]. When it comes to the safety profile of the medication, most adverse effects were mild to moderate, with 9.2% of the treatment arm experiencing serious adverse effects. A full review of the data from this study will be completed, and another phase II clinical trial using a higher dose of deucravacitinib will further evaluate its use in the management of UC [88]. The safety profile of deucravacitinib was similar to that in psoriasis patients, and no new safety signals were observed.

### 3.8. Toll-Like Receptor 9 (TLR9) Agonists

#### Cobitolimod

Cobitolimod is an oligodeoxynucleotide TLR9 agonist that induces both IL-1 and type-1 IFN expression. An effective dose of cobitolimod was tested in a double-blind, phase III, placebo-controlled trial (COLLECT) in patients with moderate-to-severe UC. In this study, two doses of topical cobitolimod 30 mg failed to meet primary and secondary endpoints at 12 weeks [89]. Higher doses of cobitolimod—31 mg, 125 mg, and 250 mg—were evaluated in another phase IIb randomized trial (CONDUCT) in patients with moderate-to-severe UC. The results showed that clinical remission occurred at week 6 in the group that received 250 mg of cobitolimod compared to the placebo group, 21% vs. 7%, respectively (*p* = 0.025) [90]. A pivotal phase III clinical trial, CONCLUDE, is underway and will test induction therapy using cobitolimod 250 mg, which was the dose that showed the best efficacy in the CONDUCT trial, and a higher dose of 500 mg [91]. Therefore, cobitolimod was well tolerated and efficacious (see Figure 1 and Figure 2).

In conclusion, the therapeutic arsenal for IBD treatment is being developed and repurposed at an accelerated pace. Several novel therapeutic targets have been identified and studied by developing small molecules and biologic agents that work on one or several combined targets. In addition to the trials that led to Food and Drug Administration (FDA) approval of certain medications, several more are currently in progress to further investigate the efficacy and safety of novel agents and should be completed within the next few years. In this overview of emerging therapies for IBD, we highlight several of these innovative medications. For instance, trials involving small-molecule drugs like Janus kinase (JAK) inhibitors and sphingosine-1-phosphate (S1P) receptor modulators have shown promising results; however, safety data, particularly the long-term risk of adverse effects, remain unclear. In addition, TYK2 and phosphodiesterase 4 inhibitors are promising classes of medications. Finally, α4 integrin and α4β7 antagonists are newer classes of medications for IBD that have passed phase 2a clinical trials.

Since these medications can be administered orally, they have the benefit of being simple to administer as compared to other IV/SC biological agents. Additionally, they do not have the drawbacks of immunogenicity and antibody development as compared to other biological therapies. Furthermore, choosing the optimal medication to treat each patient is becoming an obstacle given the wide array of therapeutic agents becoming available for the treatment of IBD. This necessitates implementing a precision medicine strategy to determine an appropriate management strategy based on patient demographics, past medical history, predictors of response, preference of administration route, and disease characteristics.

## Figures and Tables

**Figure 1 biomedicines-11-00747-f001:**
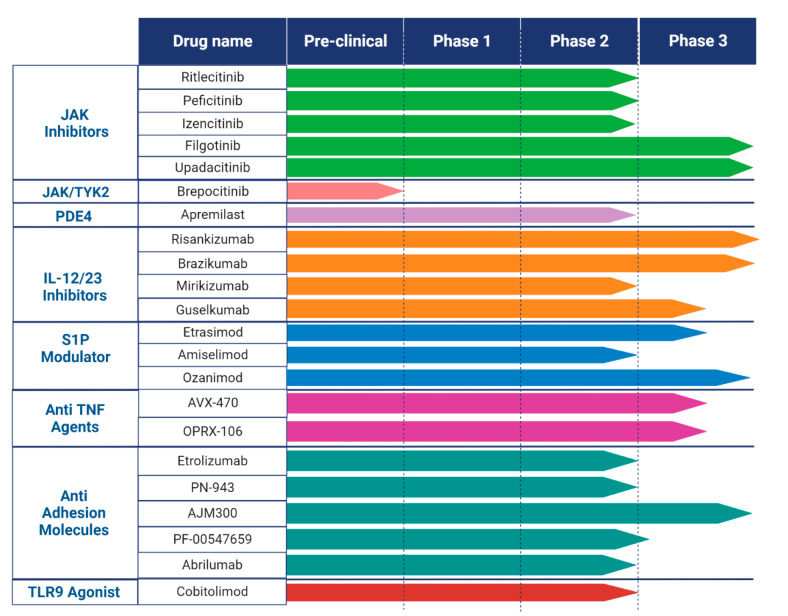
List of different biologics and small molecule drugs with their targets and current clinical trial status.

**Figure 2 biomedicines-11-00747-f002:**
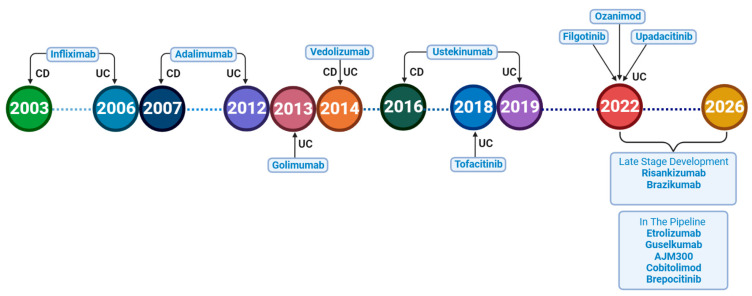
Timeline for the different biologics and small molecules approved by the FDA and in the development process for the treatment of IBD.

**Table 1 biomedicines-11-00747-t001:** Small molecules and biologics with their targets, modes of administration, and clinical trial status.

Class	Drug	Target	Route of Administration	Clinical Trial UC CD	
JAK	Tofacitinib	JAK1/JAK3	Oral	FDA Approved	Phase IIb
	Filgotinib	JAK1	Oral	Phase III	Phase III
	Upadacitinib	JAK1	Oral	FDA Approved	Phase III
	Izencitinib	JAK1	Oral	Phase IIb	
	Peficitinib	JAK1	Oral	Phase IIb	
	Ritlecitinib	JAK1	Oral	Umbrella Study	
JAK/TYK2	Brepocitinib	TYK2/JAK1	Oral	Umbrella Study	
PDE	Apremilast	PDE4	Oral	Phase II	
Anti-IL-23	Risankizumab	IL23/p19 subunit	IV or SC	Phase III	Phase III
	Brazikumab	IL23/p19 subunit	IV or SC	Phase II/OLE	Phase IIb/III
	Mirikizumab	IL23/p19 subunit	IV or SC	Phase II	
	Guselkumab	IL23/p19 subunit	IV or SC	Phase IIb/III	Phase II/III
Anti-adhesion Molecules	Etrolizumab	α4β7 integrin	SC	Phase I	Phase I
	PN-943	α4β7 integrin	Oral	Phase I	
	Vedolizumab	α4β7 integrin	IV	Phase IV	Phase IV
	AJM300	α4 integrin	Oral	Phase III	Phase III
	PF-00547659	MAdCAM	SC	Phase II completed	Phase II completed
S1P receptor modulators	Ozanimod	S1P1 and S1P5 receptors	Oral	Phase IV	Phase IV
				Phase II/III	Phase II/III
	Etrasimod	S1P1, S1P4, and S1P5 receptors	Oral	Phase II	Phase II/III
	Amiselimod	SIP, S1PR1	Oral	Phase II	Phase II
TLR9 agonist	Cobitolimod	TLR9	Topical	Phase IIb completed/Phase III under process	N/A

JAK, Janus kinase; TYK, tyrosine kinase; PDE, phosphodiesterase; IL, interleukin; MAdCAM, mucosal vascular addressin cell adhesion molecule; S1P1, sphingosine-1-phosphate; and TLR, toll-like receptor.

## Data Availability

Not applicable.

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
