# Peer review of "Exploring the Pipeline of Novel Therapies for Inflammatory Bowel Disease; State of the Art Review"

_biomedicines, 2023, doi:10.3390/biomedicines11030747_

Round 1

Reviewer 1 Report

This is a review article which summarized the latest knowledges regarding promising treatment for inflammatory bowel diseases.

The content of this article is informative and interesting for researchers and clinicians in this field.

However, the following minor issues require clarification:

1.      Please clarify the selection and exclusion criteria and the search results: how many literatures were found and how many was excluded by what reasons. Authors may add a figure of the selection tree.

2.      The mechanisms of each medication seem difficult to understand for non-specialists. Illustrated figures showing their mechanisms can help readers’ understanding.

3.      I think that the efficacy in not only induction but also maintenance treatment is an interest for clinicians. Please describe the results in each treatment from this point of view.

Author Response

Reviewer 1:

This is a review article which summarized the latest knowledges regarding promising treatment for inflammatory bowel diseases.

The content of this article is informative and interesting for researchers and clinicians in this field.

However, the following minor issues require clarification:

  1. Please clarify the selection and exclusion criteria and the search results: how many literatures were found and how many was excluded by what reasons. Authors may add a figure of the selection tree.

Thank you. As we did not do a systematic review, but a narrative review, we did not feel it was necessary to do selection and exclusion criteria as well as a PRISMA flow chart. Below are examples of narrative reviews:

https://www.nejm.org/doi/10.1056/NEJMra1914900

https://www.nature.com/articles/s41575-022-00658-y

https://www.gastroenterologyandhepatology.net/archives/august-2022/novel-therapies-for-patients-with-inflammatory-bowel-disease/

  1. The mechanisms of each medication seem difficult to understand for non-specialists. Illustrated figures showing their mechanisms can help readers’ understanding.

Thank you for the great idea. We made a mechanism of action figure and will be presented as a visual abstract.

  1. I think that the efficacy in not only induction but also maintenance treatment is an interest for clinicians. Please describe the results in each treatment from this point of view.

Thank you. We discussed maintenance therapy whereever the study or trial involved maintenance arm. For those studies where only induction therapy were published, we did not discuss maintenance.

Reviewer 2 Report

In this review paper, the authors have described the therapeutic agents that are in phase II, phase III development and those which are in clinical trials for the potential treatment of inflammatory Bowel disease (IBD). The discussed drugs include  sphingosine -1-phosphate receptor modulators, selective Janus kinase inhibitors and anti-interleukins as well as other small molecules (e.g., AVX-470, OPRX-106PN-943 etc.,).  Overall, it is a well written and informative review paper.    

Author Response

In this review paper, the authors have described the therapeutic agents that are in phase II, phase III development and those which are in clinical trials for the potential treatment of inflammatory Bowel disease (IBD). The discussed drugs include  sphingosine -1-phosphate receptor modulators, selective Janus kinase inhibitors and anti-interleukins as well as other small molecules (e.g., AVX-470, OPRX-106PN-943 etc.,).  Overall, it is a well written and informative review paper.    

Thank you.

Reviewer 3 Report

Well done and organized.

This manuscript reviews the literature regarding treatment options for inflammatory bowel and Crohn’s diseases.  These are major diseases world-wide and this is an informatory article on this important topic.

The manuscript is well written and documented.

The only minor concern is that the authors should at least briefly mention the advent of cellular therapies in the conclusion.

Author Response

Well done and organized.

This manuscript reviews the literature regarding treatment options for inflammatory bowel and Crohn’s diseases.  These are major diseases world-wide and this is an informatory article on this important topic.

The manuscript is well written and documented.

The only minor concern is that the authors should at least briefly mention the advent of cellular therapies in the conclusion.

Thank you. Added
